# *In Vitro* Screening of Endophytic *Micromonospora* Strains Associated with White Clover for Antimicrobial Activity against Phytopathogenic Fungi and Promotion of Plant Growth

Wojciech Sokołowski *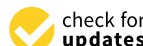, Sylwia Wdowiak-Wróbel *, Monika Marek-Kozaczuk and Michał Kalita

Department of Genetics and Microbiology, Institute of Biological Sciences, University of Maria Curie-Skłodowska, Akademicka 19, 20-033 Lublin, Poland; monika.marek-kozaczuk@mail.umcs.pl (M.M.-K.); michal.kalita@mail.umcs.pl (M.K.)
* Correspondence: wojciech.v.sokolowski@gmail.com (W.S.); sylwia.wdowiak-wrobel@mail.umcs.pl (S.W.-W.)

**Abstract:** Bacteria belonging to the genus *Micromonospora* are recognized as microorganisms with the potential to be used in biotechnology processes, given their beneficial influence on plant growth and the biocontrol of phytopathogens. In this study, nineteen *Micromonospora* isolates originating from the root nodules of white clover plants were taxonomically assigned based on the phylogenetic analysis of the 16S rRNA gene and four housekeeping genes. The antifungal properties of the bacteria against phytopathogenic *Botrytis cinerea*, *Fusarium oxysporum*, *Fusarium equiseti*, *Sclerotinia sclerotiorum*, and *Verticillium albo-atrum* were tested with the agar plug test and the dual culture test. The ability to produce various metallophores was determined with the agar plug diffusion test on modified chrome azurol S (CAS) agar medium. International *Streptomyces* Project-2 medium (ISP2) broth amended with 0.2% L-tryptophan was used to indicate the bacterial ability to produce auxins. The strains belonging to *M. tulbaghiae*, *M. inaquosa*, and *M. violae* showed *in vitro* potential as antimicrobial agents against the tested fungi. *M. inaquosa* strain 152, *M. violae* strain 126, *M. violae* strain 66, and *M. violae* strain 45 were recognized as the most efficient metallophore producers. *M. alfalfae* strain 55 and *M. lupini* strain 5052 were identified as the most promising auxin compound producers and, therefore, show potential as plant-growth-promoting bacteria.

**Keywords:** *Micromonospora*; plant growth promotion; biological control; endophytes; phytopathogenic fungi; multilocus sequence analysis

## 1. Introduction

The genus *Micromonospora* (phylum: *Actinobacteria*) was first described by Ørskov in 1926 and currently includes 122 validly published species ("https://lpsn.dsmz.de/genus/micromonospora" accessed on 15 March 2024). These Gram-positive, aerobic, and filamentous bacteria are characterized by a well-developed substrate mycelium. The bacteria of *Micromonospora* genus are widespread in the environment, e.g., the soil, plant internal tissues, marine sediments, desert sands, and even marine sponges, rocks, and Antarctic sands [1–3]. *Micromonospora* members are regarded as rich biosources of novel bioactive compounds with promising uses in medicine due to their ability to produce antibiotics (probably the second antibiotic producer after *Streptomyces*), antitumor substances, enzyme inhibitors, and antioxidants [4–6]. Moreover, the biochemical properties of *Micromonospora* show its potential for use in various biotechnological processes, such as nitrogen preservation in composting [7], the degradation of organic and artificial polymers [8–10], and even in the concrete industry as a biohealing agent [3]. The beneficial influence of these bacteria on soil ecology and plant growth and development has recently been elucidated [5]. Endophytic *Micromonospora* isolates, successfully recovered from different parts of various plants including the roots, leaves, or root nodules of legume and actinorhizal plants [11], have been shown to be beneficial for plants, with an important role as plant-growth-promoting

bacteria (PGPB) and biocontrol agents [6,12]. Many of them have been recovered and investigated in the last decade [13,14]. In this study, nineteen endophytic bacterial isolates originating from the root nodules of white clover were recovered and classified into the genus *Micromonospora*. Their antifungal properties against common phytopathogens were examined. Their ability to produce bacterial metallophores and indole-3-acetic acid was tested to assess the function of the isolates as PGPB.

## 2. Materials and Methods

### 2.1. Plant Material and Isolation of Micromonospora Strains

Bacteria were isolated from root nodules of wild white clover plants (*Trifolium repens* L.) collected in Poland and South Africa. Until bacterial isolation, the nodules were stored at 4 °C. Following the procedure developed by Vincent [15], the root nodules were washed in running tap water and sterilized by soaking in a 0.1% $HgCl_2$ (Pol-aura, Warszawa, Poland) solution for 1 min and 75% EtOH for 1 min, followed by rinsing five times in sterile deionized water. Surface-sterilized nodules were crushed aseptically, streaked onto 79CA agar medium, and incubated at 28 °C for 35 days. Orange-to-brown *Micromonospora*-like colonies were transferred and cultivated on ISP2 agar [16] and used in further analysis. Two isolates obtained from plants collected in Poland (strains 5052 and 5056) and seventeen other isolates obtained from African plants (Supplementary Material, Table S1) were examined. All isolates were deposited with the Department of Genetic and Microbiology (UMCS, Lublin, Poland) microbial culture collections and preserved in 15% glycerol (Pol-aura, Warszawa, Poland) ($v/v$) at −70 °C.

### 2.2. Genomic DNA Extraction

Extraction of genomic DNA from the endophytic bacterial isolates was conducted following the method proposed by Pitcher et al. [17], with modification of the bacterial cell degradation step. The 15 min incubation of the bacterial suspension in Tris-EDTA (TE) buffer with guanidinium thiocyanate-EDTA-sarcosyl (GES) reagent was replaced by 16 h incubation at 37 °C. Instead of GES buffer, 20 μL of 228 U/μL lysozyme (A&A Biotechnology, Gdansk, Poland) and 10 μL of 10 U/μL mutanolysin (A&A Biotechnology, Gdansk, Poland) were added.

### 2.3. Amplification of 16S rRNA Gene and Housekeeping Genes

Amplification of 16S rRNA gene fragments (about 1500 bp) was carried out by PCR using fD1d (5′-GAGAGTTTGATCCTGGCTCAGA-3′) and rPla (5′-CTACGGCTACCTTGTTA CGACTT-3′) primers [18]. The reaction mixture contained 12.5 μL of Taq PCR Master Mix 2× (Eurx, Gdańsk, Poland), 0.25 μL of each primer (100 mM), 100 ng of DNA, and up to 25 μL of nuclease-free sterile water. The initial denaturation step of PCR was performed at 95 °C for 4 min, followed by 34 cycles of denaturation at 95 °C for 45 s, annealing at 55 °C for 1 min, extension at 72 °C for 2 min, and final extension at 72 °C for 7 min. Housekeeping genes (*recA*, *gyrB*, *rpoB*, and *atpD*) were amplified according to previously published PCR conditions and primers [19]. The amplicons were analyzed by electrophoresis on 1% agarose (Maximus, Łódź, Poland) gel. Amplified gene fragments were purified with a Clean-Up Kit (A&A Biotechnology, Gdansk, Poland) according to the manufacturer's instructions and sequenced by Genomed S. A. Company (Warsaw, Poland). DNA sequences of amplified genes were deposited with GenBank under accession numbers OQ892262-OQ892276, OQ880495-OQ880497, and OQ874508 for the 16S rRNA gene and OQ943085-OQ943103 (*recA*), OQ943104-OQ943122 (*gyrB*), OQ943123-OQ943141 (*rpoB*), and OQ943142-OQ943160 (*atpD*) for the housekeeping genes.

### 2.4. Phylogenetic Analysis of Sequencing Data

The most closely related sequences to the 16S rRNA, *recA*, *gyrB*, *rpoB*, and *atpD* genes of the tested isolates were obtained from the GenBank database with the Basic Local Alignment Search Tool (BLAST) algorithm [20]. MEGA software (version 7) [21] was used to

construct the alignments of the 16S rRNA gene and concatenated housekeeping gene sequences. In the next step, phylogenetic trees were constructed according to the models of evolution determined by the model test (MEGA 7 software option). The maximum likelihood algorithm evaluated by bootstrap analysis of 1000 replicates was used. *Catellatospora citrea* DSM 44097 was used as an outgroup in both phylogenetic analyses.

### 2.5. Evaluation of Antifungal Properties

The antifungal activity of the tested bacteria was evaluated *in vitro* with the agar plug diffusion test [22] and the dual culture plate assay according to the method developed by Trujillo et al. [1]. Five fungal plant pathogens, *Sclerotinia sclerotiorum* strain 10Ss01, *Fusarium oxysporum* strain 10Fo01, *Botrytis cinerea* strain 10Bc01, *Verticillium albo-atrum* CBS 745.83, and a wild pathogenic *Fusarium equiseti* isolate, were used in both methods. Phytopathogenic fungi were obtained from the Institute of Horticulture in Skierniewice, Poland, and the University of Life Sciences in Lublin, Poland.

### 2.6. Agar Plug Diffusion Assay

Seven-day-old cultures of the tested bacteria growing on ISP2 agar were used to inoculate different types of solid culture media dedicated to the investigation of actinobacterial growth and antagonism (ISP2, ISP3, ISP4, and ISP5) [16], minimal medium [23] (with addition of 10 g/L of mannitol instead of glucose), and yeast extract–malt extract (YEME) agar [24]. After 11 days of incubation at 28 °C, 0.7 cm diameter agar plugs were cut with a sterile cork-borer and placed on the borders of Petri dishes (four plugs per plate) with potato dextrose agar (PDA) [25]. In the center of the PDA plates, 0.7 cm diameter plugs of the 7-day-old phytopathogenic fungal cultures were placed simultaneously and incubated at 28 °C. Plates without agar plugs with the bacteria, inoculated only with the fungal strain, were considered as a negative control. The inhibition zones of fungal growth around the agar plugs indicated bacterial abilities to suppress the phytopathogenic fungi; the measurement of the inhibition range was carried out after the full colonization of the control plates by the phytopathogenic fungi. Biological triplicates were performed for each treatment.

### 2.7. Dual Culture Plate Assay

Seven-day-old actinobacteria cultured on ISP2 medium were suspended in sterile saline (0.85% NaCl solution) at optical density (OD$_{600}$) = 0.2. Then, 15 µL of bacterial suspension was inoculated on the edge of Petri dishes with SA1 medium [1] and incubated for 7 days at 28 °C. After the incubation period, 0.7 cm diameter agar plugs of the phytopathogenic fungi were placed in the center of Petri dishes and incubated at a suitable temperature to the moment of the full colonization by the fungi on the control plates (fungi without bacteria). After this period, the inhibition zones of fungal growth around the bacterial colonies were measured. In the case of *V. albo-atrum*, suppression range was measured after 28 days of fungal growth. Biological triplicates were performed for each tested strain.

### 2.8. Production of Metallophores and Indole-3-Acetic Acid

The potential to produce iron, copper, aluminum, and arsenic-sequestering metallophores was tested on chrome azurol S (CAS) agar, where iron ions were substituted with the appropriate metal ions, according to the modification described by Mehnert et al. [26]. The ability to produce metallophores was determined following the method proposed by Rungin et al. [27]. For this, 0.7 cm diameter agar plugs cut from 7-day-old bacterial cultures on ISP2 medium agar were placed on CAS medium agar with the addition of the appropriate metal compounds and incubated at 28 °C for 14 days. Biological triplicates were prepared for each treatment; sterile ISP2 agar plugs were used as a negative control (Supplementary Material, Figure S2). An orange halo around the agar plug was considered a positive result. The indole-3-acetic acid (IAA) produced by the tested isolates was assessed using a colorimetric assay [28]. For this, 100 µL of the bacterial suspension

($OD_{600}$ = 0.2) was inoculated into 5 mL of ISP2 broth amended with 0.2% L-tryptophan (Pol-aura, Warszawa, Poland). After 14 days of agitation at 28 °C and 160 rpm, 1 mL of each bacterial culture was centrifuged at 11,000 rpm for 5 min, and 100 µL of the supernatant was mixed with 200 µL of Salkowski reagent [29]. After 25 min of incubation in the dark, the samples were measured at 530 nm with a Biotek Synergy H1 Hybrid Microplate Reader spectrophotometer. The IAA production was estimated by comparison of the obtained results with the prepared IAA standard curve (range of 10–100 µg × $mL^{-1}$ ISP2 broth). Biological triplicates were performed for each treatment.

## 3. Results and Discussion

### 3.1. Isolation of Micromonospora Strains and Genomic DNA Extraction

79CA plates inoculated with aseptically crushed root nodules slurry were incubated at 28 °C for 35 days, and the newly grown, gradually appearing small orange colonies were subcultured. The genomic DNA of pure colonies, extracted with the modified Pitcher method [17], was used for genotyping of all isolates using the BOX-PCR method. Based on the BOX-PCR genotyping, nineteen different isolates were selected for genetic analyses and further tests.

### 3.2. Phylogenetic Analysis of 16S rRNA Gene Sequences

The evolutionary phylogenetic tree of the 16S rRNA gene, based on the 1274 bp sequences of the tested isolates and the most closely related reference strains obtained from the GenBank database, is presented in Supplementary Material Figure S1. The analysis showed the segregation of the taxa into two large groups comprising 15 and 4 isolates, both with two smaller subgroups. All the tested strains showed high similarities of the 16S rRNA gene sequences with various *Micromonospora* species (from 99.6 to 100% similarity). These results allowed the classification of the isolates to the *Micromonospora* genus, but the limitation of the resolution power of this method did not allow affiliating the isolates to the species level [30,31].

### 3.3. Multilocus Sequence Analysis (MLSA) of Housekeeping Genes

To reveal the relationship between the tested isolates and the most related reference strains at the species level, multilocus sequence analysis (MLSA) was carried out [31,32]. As a result, the maximum likelihood phylogenetic tree of the strains examined in this study and the valid taxa based on the concatenated 2620 bp of four housekeeping gene sequence fragments is presented in Figure 1. The tested strains were divided into two large groups (encompassing 15 and 4 strains) in a similar way as in the 16S rRNA gene phylogenetic tree. Strains 43, 249, 126, 29, 45, and 66 were grouped with the most closely related *Micromonospora violae* DSM 45888 (sequence similarity 97.4%). The second phylogenetic group comprised 152, 33, N2, 230, 30, and *Micromonospora inaquosa* LB39 as the most closely related reference species, with a sequence similarity range from 97.3 to 97.8%. *Micromonospora purpureochromogenes* DSM 43821 was the most closely related to strains 5056 and N5 (97.6–98.8% sequence similarity); strains 48 and N4 were closely related (99.1–99.5% sequence similarity) to *Micromonospora tulbaghiae* DSM 45142. Strains 53, 55, and 50 showed 99.1% sequence similarity to reference strain *Micromonospora alfalfae* MED01. Strain 5052 exhibitsed 98.5% sequence similarity with strain *Micromonospora lupini* DSM 44874.

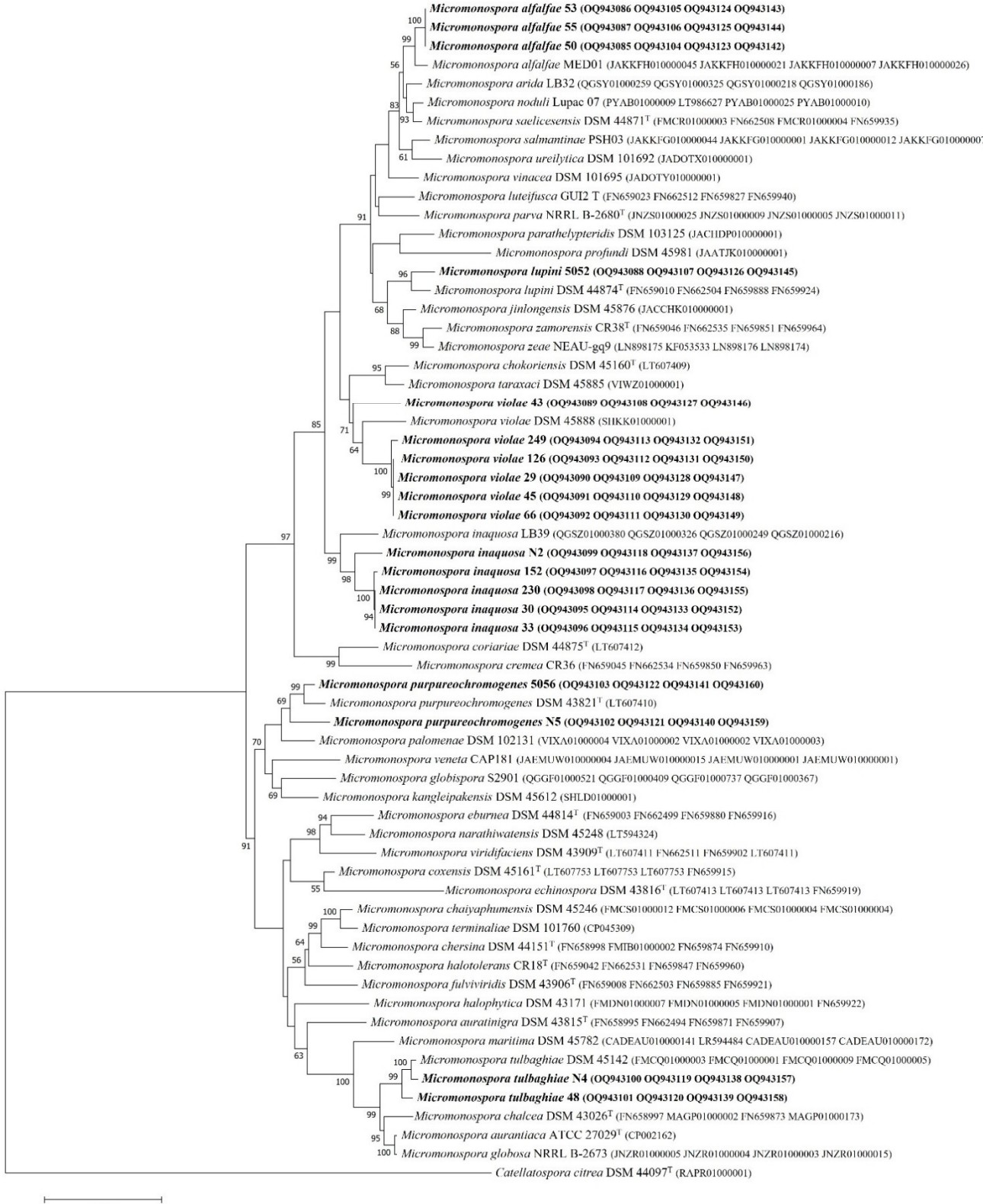

**Figure 1.** Phylogenetic MLSA tree constructed using the general time-reversible model [33]. Discrete gamma (+G) distribution was used to model evolutionary rate differences among sites, and the rate variation model allowed some sites to be evolutionarily invariant (+I). Bootstrap values [%] higher than 50 are shown next to the branches. The scale bar represents 0.05 substitutions per nucleotide position. Bold font indicates tested bacterial strains.

*3.4. Antifungal Properties*

The evaluation of the antifungal properties of endophytic *Micromonospora* was the main object of this study. The bacterial ability to suppress phytopathogenic fungi was firstly assessed by the agar plug diffusion test with six various culture media used for bacterial growth. Despite this, the suppression of plant pathogens was not observed. In the second step, the dual culture plate assay was carried out and was recognized as appropriate tool for detecting the antifungal properties of the tested endophytic strains. The suppression range of phytopathogenic fungi was measured after 4 days of incubation (*B. cinerea* and *S. sclerotiorum*), 6 days of incubation (*F. oxysporum* and *F. equiseti*), or 28 days of incubation (*V. albo-atrum*). An antimicrobial effect against the phytopathogenic fungi was observed (Figure 2).

| | *S. sclerotiorum* | *B. cinerea* | *F. oxysporum* | *F. equiseti* | *V. albo-atrum* |
|---|---|---|---|---|---|
| *M. inaquosa 30* | - | - | - | - | 1 |
| *M. inaquosa 152* | - | - | 3 | 2 | - |
| *M. tulbaghiae 48* | - | - | 3 | 3 | 3 |
| *M. violae 29* | - | - | - | 3 | - |
| *M. violae 66* | - | 3 | 4 | 3 | - |

**Figure 2.** Distances between bacterial and fungal colonies [mm] in dual culture plate assay, indicating ability of *Micromonospora* strains to inhibit mycelial growth of phytopathogenic fungi *in vitro*. Inhibition distances were calculated from triplicates on the basis of the arithmetic mean.

Five strains (~26% of the total) showed the ability to suppress phytopathogenic fungi; the diameter of the inhibition zone ranged from 1 mm to 4 mm. *Micromonospora tulbaghiae* 48 showed antagonism against *F. oxysporum*, *F. equiseti* (Figure 3a), and *V. albo-atrum* (Figure 3b). *Micromonospora violae* 66 was able to suppress three fungal pathogens (*B. cinerea*—Figure 3c, *F. oxysporum* and *F. equiseti*); *Micromonospora inaquosa* 152 showed antifungal properties against *F. oxysporum* and *F. equiseti* (Figure 3d); *Micromonospora inaquosa* 30 showed antimicrobial activity against *V. albo-atrum*. Strain *Micromonospora violae* 29 was able to inhibit the growth of *F. equiseti*. Members of the *Micromonospora* genus are considered great candidates as novel bioactive compound producers, including for the production of secondary metabolites [34]. The results confirm those of previous reports, where *Micromonospora* strains have shown antagonistic activity against other microorganisms, including fungi [35–37].

However, in our study, the antifungal activity was observed only in the dual culture plate assay, where SA1 agar was used as a growth medium. It is worth emphasizing that the production of antimicrobial compounds by actinobacteria is highly dependent on growth conditions, including carbon and nitrogen sources [38], and various strains of *Micromonospora* may require differently composed media to produce antimicrobials [39]. SA1 agar was the only medium to accurately indicate the antimicrobial activity of the tested strains. Similarly, SA1 agar was also successfully used by Martínez-Hidalgo et al. [40], who evaluated the antimicrobial activity of *Micromonospora* strains against fungal plant pathogens, including *Sclerotinia sclerotiorum* and *Botrytis cinerea*. As reported by Amin et al. [38], the addition of glucose (the main carbon source in SA1 medium) resulted in an increase in the antimicrobial agent yields of *Micromonospora*. In turn, Mark et al. [41] reported the antimicrobial activity of *Micromonospora* bacteria with properties inducible also by culturing on International *Streptomyces* Project (ISP) media, which proved to be unhelpful in our investigation.

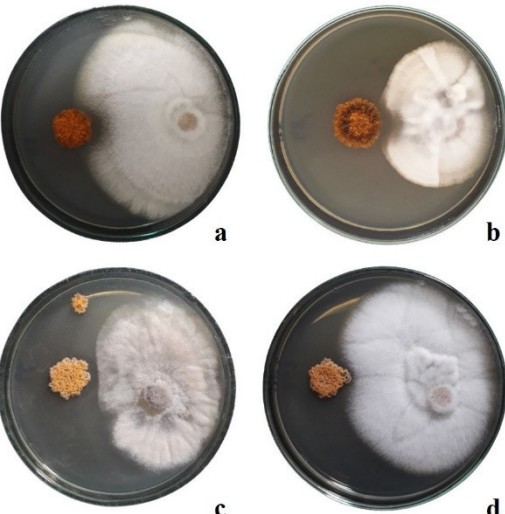

**Figure 3.** Zone of suppression of fungal growth, evaluated by dual culture plate assay. On the left side, the orange colonies of the tested bacteria against phytopathogenic fungi mycelium on the right side: (**a**) *M. tulbaghiae* 48 against *F. equiseti*; (**b**) *M. tulbaghiae* 48 against *V. albo-atrum*; (**c**) *M. violae* 66 against *B. cinerea*; (**d**) *M. inaquosa* 152 against *F. equiseti*.

### 3.5. Production of Metallophores

Orange halos around the agar plugs (Figure 4) were observed in the samples with all nineteen tested strains. This indicated their ability to produce metallophores and sequester ions of iron (Fe), copper (Cu), aluminum (Al), and arsenic (As). The halo zone diameter ranged from 16 mm to 40 mm (Figure 5). The largest siderophore diffusion zones (over 30 mm for three different metal ions added to the agar) were observed for the strains *M. inaquosa* 152, *M. violae* 126, *M. violae* 66, and *M. violae* 45.

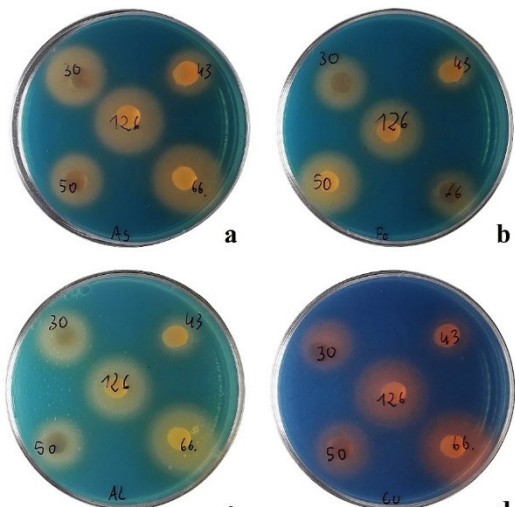

**Figure 4.** Orange halos around agar plugs cut from 7-day-old bacteria cultures on ISP2 agar. The results indicated the ability of *M. inaquosa* 30, *M. violae* 43, *M. alfalfae* 50, *M. violae* 66, and *M. violae* 126 to produce metallophores able to sequester metal ions from various types of chrome azurol S agar, with the addition of ions: (**a**) arsenic (As); (**b**) iron (Fe); (**c**) aluminum (Al); (**d**) copper (Cu). The numbers shown in the center of the halo zones refer to the designations of the tested bacterial strains (Table S1, Supplementary Information).

| | Fe-CAS | Cu-CAS | Al-CAS | As-CAS |
|---|---|---|---|---|
| *M. alfalfae 50* | 26 (3.9) | 27 (1.0) | 21 (0.0) | 28 (0.3) |
| *M. alfalfae 53* | 20 (0.4) | 25 (0.6) | 23 (0.4) | 36 (0.5) |
| *M. alfalfae 55* | 24 (1.2) | 29 (0.4) | 25 (0.8) | 34 (1.0) |
| *M. inaquosa 30* | 23 (0.0) | 24 (2.0) | 20 (4.1) | 28 (4.7) |
| *M. inaquosa 33* | 19 (1.5) | 20 (1.8) | 19 (0.0) | 25 (1.5) |
| *M. inaquosa 152* | 18 (0.5) | 33 (1.5) | 33 (0.8) | 40 (0.5) |
| *M. inaquosa 230* | 27 (1.0) | 32 (1.6) | 28 (0.2) | 37 (1.3) |
| *M. inaquosa N2* | 27 (0.8) | 31 (1.3) | 26 (0.2) | 38 (0.8) |
| *M. lupini 5052* | 22 (0.3) | 27 (1.4) | 27 (2.3) | 32 (5.0) |
| *M. purpureochromogenes N5* | 15 (1.2) | 17 (0.3) | 16 (0.6) | 22 (0.0) |
| *M. purpureochromogenes 5056* | 21 (0.1) | 21 (1.1) | 21 (1.0) | 30 (2.0) |
| *M. tulbaghiae 48* | 17 (0.9) | 19 (0.8) | 18 (0.3) | 25 (1.1) |
| *M. tulbaghiae N4* | 15 (1.3) | 18 (0.8) | 15 (0.8) | 21 (0.3) |
| *M. violae 29* | 27 (1.6) | 29 (0.6) | 25 (1.0) | 36 (2.0) |
| *M. violae 43* | 19 (2.2) | 16 (2.2) | 17 (2.2) | 23 (1.3) |
| *M. violae 45* | 25 (0.1) | 32 (0.2) | 32 (1.3) | 36 (1.8) |
| *M. violae 66* | 27 (3.3) | 37 (1.2) | 40 (0.5) | 31 (0.0) |
| *M. violae 126* | 30 (0.4) | 36 (1.8) | 27 (1.0) | 40 (0.5) |
| *M. violae 249* | 23 (0.5) | 27 (3.8) | 25 (0.0) | 32 (2.0) |

**Figure 5.** The ability of *Micromonospora* strains to produce various metallophores was evaluated on chrome azurol S agar supplemented with ions of iron (Fe), copper (Cu), aluminum (Al), and arsenic (As). The halo zones around agar plugs cut from 7-day-old bacteria cultures (on ISP2 agar) were measured after 14 days of incubation at 28 °C. The average diameter of halos [mm] was calculated using the arithmetic mean from triplicates. Values in brackets represent standard deviation.

The smallest zones were observed for strains *M. violae* 43, *M. tulbaghiae* 48, *M. tulbaghiae* N4 and *Micromonospora purpureochromogenes* N5 (under 20 mm for three of four tested metals). The halo zones on the As-CAS medium had the largest diameters. Due to the high toxicity of hexadecyltrimethylammonium bromide (HDTMA), it was not possible to grow the tested actinomycetes on CAS agar. Metallophores, i.e., various ion-chelating molecules, facilitate metal uptake as nutrient elements and are useful as metal stress management agents. This study found that all the tested *Micromonospora* strains exhibited the ability to produce various metallophores. The same conclusion was reported in previous *in vitro* and *in vivo* studies conducted by Ortúzar et al. [42]. Moreover, the genome mining of the complete genome sequence of *Micromonospora craniellae* LHW63014T [43] showed biosynthetic gene clusters, indicating the ability to produce ion-chelating agents.

*3.6. Production of Indole-3-Acetic Acid*

The spectrophotometric comparison of the standard curve of IAA and that of the samples of bacterial supernatant and Salkowsky reagent indicated the ability of all the studied strains of *Micromonospora* to produce IAA as a result of tryptophan metabolism (Figure 6). *M. alfalfae* strain 55, *M. purpureochromogenes* strain N5, *M. lupini* strain 5052, and *M. violae* strain 43 showed the highest productivity (above 27 μg/mL). The lowest amount of IAA was detected for *M. purpureochromogenes* strain 5056.

The production of phytohormones plays a significant role in plant growth promotion by plant-beneficial bacteria [44]. As demonstrated by the widely used spectrophotometric measurement of IAA production (despite the limitation of this method described by Goswami et al. [45]), numerous *Micromonospora* isolates have been reported as auxin-producing PGPB [46–48].

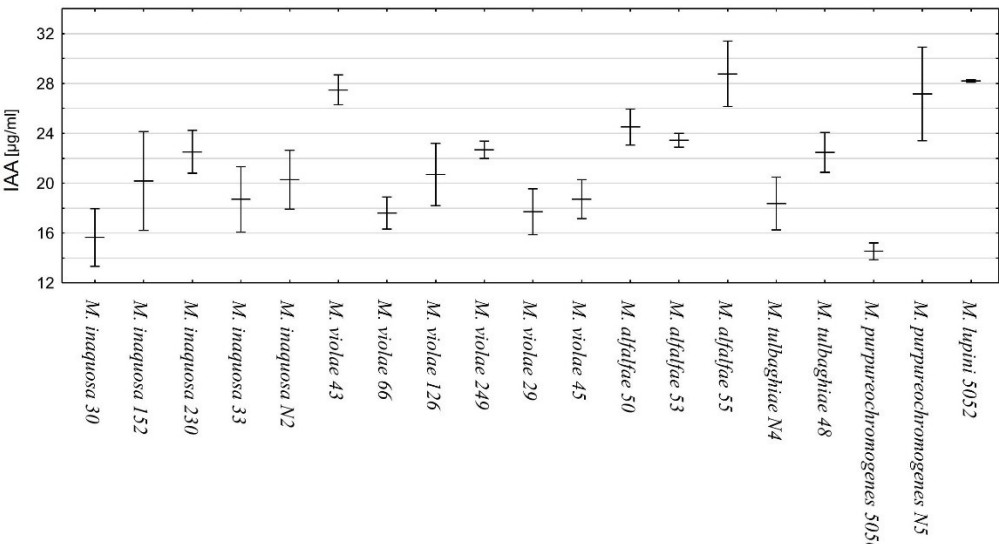

**Figure 6.** Ability of *Micromonospora* strains to produce indole-3-acetic acid (IAA). The center line reveals the results of the arithmetic mean calculated from triplicates; the whiskers reveal the standard deviation.

## 4. Conclusions

Nineteen endophytic *Micromonospora* strains isolated from the root nodules of white clover (*Trifolium repens* L.) were classified as *M. inaquosa*, *M. violae*, *M. alfalfae*, *M. tulbaghiae*, *M. purpureochromogenes*, and *M. lupini*. The antifungal properties of and two mechanisms of plant growth promotion (production of metallophores and auxin) by these strains were assessed. The strains *M. tulbaghiae* 48, *M. inaquosa* 30, *M. inaquosa* 152, *M. violae* 66, and *M. violae* 29 showed antimicrobial properties against phytopathogenic fungi, especially against *Fusarium equiseti* and *Fusarium oxysporum*. Further comprehensive analyses of these phenomena are necessary, especially the characterization of antifungal bacterial metabolites. Moreover, strains with the ability to produce metallophores and auxin are candidates for use in agriculture to promote plant growth, which should be confirmed by in planta studies. The most promising strains seem to be *M. alfalfae* 55 and *M. lupini* 5052. The results indicate the *in vitro* potential of the selected strains as plant growth promotion agents. Further studies are necessary to evaluate the use of the studied strains in the biocontrol of phytopathogenic fungi.

**Supplementary Materials:** The following supporting information can be downloaded at: https://www.mdpi.com/article/10.3390/agronomy14051062/s1. Table S1. *Micromonospora* strains used in this study. Figure S1. The maximum-likelihood phylogenetic tree of 16S rRNA gene fragments of the nineteen endophytic isolates and fourteen most closely related taxa was constructed using the Tamura-Nei model [49]. Discrete Gamma (+G) distribution was used to model evolutionary rate differences among sites and the rate variation model allowed some sites to be evolutionarily invariant (+I). Bootstrap values [%] higher than 50 are shown next to the branches. The scale bar represents 0.005 substitutions per nucleotide position. Figure S2. Production of metallophores-sterile ISP2 agar plugs placed in the centre were used as a negative control (no halo).

**Author Contributions:** W.S., S.W.-W., M.M.-K. and M.K. participated in the conceptualization, design, and modification of methods, data analysis, and manuscript writing. Moreover, the biological assays and data collection were performed by W.S. All authors have read and agreed to the published version of the manuscript.

**Funding:** This research received no external funding.

**Data Availability Statement:** The gene sequences data presented in this study are available in GenBank repository at "https://www.ncbi.nlm.nih.gov/genbank" accessed on 10 March 2024; the reference numbers of the sequences are cited in the main text.

**Conflicts of Interest:** The authors declare no conflicts of interest. The funders had no role in the design of the study; in the collection, analyses, or interpretation of data; in the writing of the manuscript; or in the decision to publish the results.

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
