# Peer review of "In Vitro Screening of Endophytic Micromonospora Strains Associated with White Clover for Antimicrobial Activity against Phytopathogenic Fungi and Promotion of Plant Growth"

_agronomy, doi:10.3390/agronomy14051062_

Round 1

Reviewer 1 Report

Comments and Suggestions for Authors

In this work, authors isolated and identified nineteen Micromonospora isolates. Authors also tested the antifungal properties of the bacteria against phytopathogenic Botrytis

cinerea, Fusarium oxysporum, Fusarium equiseti, Sclerotinia sclerotiorum, and Verticillium alboatrum in vitro. The results showed that some strains showed potential as antimicrobial agents against fungi and some strains had promoting plant growth. However, I report some points that should be implemented.

1. Figure 3 and Figure 6 should be revised. It is better to display quantitative data.

2. CAS and ISP2 in Abstract should be displayed the full name.

3. Figure 1 may be deleted or as a supplementary figure.

Author Response

We would like to thank for the constructive comments on our manuscript. According to your suggestions, Figure 1 has been placed in the Supplementary Materials. Figure 3 and Figure 6 (now Figure 2 and Figure 5, respectively) have been revised. The quantitative data of fungal inhibition ranges and metallophores-related halo zones have been added. The abbreviations CAS and ISP have been explained in the abstract.

Reviewer 2 Report

Comments and Suggestions for Authors

This manuscript investigates nineteen Micromonospora strains isolated from the root nodules of white clover plants. It assesses their potential as biological control agents against phytopathogenic fungi and as enhancers of plant growth. The research evaluates the antifungal properties of these strains using in vitro tests and examines their capacity to produce substances that promote plant growth, such as metallophores and auxins. The paper is well-written in all its sections.

The only concern is related to the orange halos around agar plugs that are shown in Figure 5. This figure is very confusing in terms of estimating the halo zone diameters around each strain. I recommend adding a separate table listing all 9 strains with the corresponding halo zone diameters in mm. I also prefer to see all these agar plates, without color effects, added to the paper’s supplementary materials.  

Author Response

We would like to thank for your review. According to your commentaries, the quantitative data of halo zones diameters have been added to Figure 6 (now Figure 5). Moreover, the images of control plates with sterile agar plugs (without halo zones) have been added to the Supplementary Materials as Figure S2.

Reviewer 3 Report

Comments and Suggestions for Authors

The manuscript by Sokolowski and colleagues presents a study of the biocontrol ability of 19 Micromonospora strains isolated in Poland and South Africa.

The manuscript is incoherent in its content and lacking originality and interest. Therefore, I have to propose to reject the manuscript. 

After a correct taxonomical description and classification of their genetic relationship, the authors question the strains ability to inhibit fungal growth. Firstly, they measure fungal growth reduction using dual cultivar plate assays. In a second step, the production of metallophores and Indole-3-acetic acids has been analysed. 

The authors have used modern, state of the art methodology for all their analysis and conclude correctly in the discussion part (yet, results and discussion) and in the conclusions.

My concern with this manuscript is, that all information generated is already available in the current literature. Most of it has been cited by the authors themselves. They just extend the trials to a, seemingly, new set of 19 strains isolated from white clover in Poland and South Africa. 

- Nothing is explained on these 19 strains. Why have they been included in this trial? Was it biotic or abiotic resistance of the original clover? Adaptation to local conditions?  or just by chance? 

- Arguably, the prove that certain strains have a fungistatic action in dual culture assays is interesting but not a new finding in the genus Micromonospora spp. It is intriguing that the substrate determines the fungistatic action; indeed, it works only on SA1 but not on other tested substrates. In the discussion, the authors try to find an explanation, yet the quiry remains at a rather shallow level. I would like to stress that such an in vitro test may give a hint towards the biochemical capacities of microbes to inhibit or interact with plant pathogens, yet it is by far not a prove for their applicability in agriculture or horticultre.  A prove can only be obtained in a correctly set up plant-pathogen biotest.

My concern here is, that the results cannot be transferred to any agricultural reality.  I conclude, that the proposed title of the manuscript is therefore misleading since the article provides no prove that any of the strains may give a benefit in agriculture.   

Some points, as they come:

- when using mercury for disinfection, how do you handle the waste? Today, we use javel water or simply 70% ethanol.

- The authors have measured the inhibition zones  and metallophore-activity zones around the agar plugs. Have the authors compared the zones between strains and between treatments statistically? 

Comments on the Quality of English Language

The English used in the manuscript seems all right to me. No major problems.

Author Response

Nothing is explained on these 19 strains. Why have they been included in this trial? Was it biotic or abiotic resistance of the original clover? Adaptation to local conditions? or just by chan-ce?

In 2019-2022, we participated in the Polish-South African project "Composition and function of the microbiome of root nodules of Trifolium rubens and Trifolium africanum" (PL-RPA2/07/TRIFOMIKRO/2019). Among various bacteria isolated from the root nodules of clover growing in Poland and South Africa, we identified several strains of the Micromonospora genus. We agree with Reviewer 3 that there are many published reports indicating that actinomycetes constitute a group of microorganisms with great potential for use in various industries (e.g. pharmaceutical, agriculture). Therefore, we decided to test our strains for potential growth promotion mechanisms and antifungal properties. The results presented in the submitted article are part of the doctoral dissertation written by PhD student Wojciech Sokołowski.

Arguably, the prove that certain strains have a fungistatic action in dual culture assays is interesting but not a new finding in the genus Micromonospora spp. It is intriguing that the substrate determines the fungistatic action; indeed, it works only on SA1 but not on other tested substrates. In the discussion, the authors try to find an explanation, yet the quiry remains at a rather shallow level. I would like to stress that such an in vitro test may give a hint towards the biochemical capacities of microbes to inhibit or interact with plant pathogens, yet it is by far not a prove for their applicability in agriculture or horticultre. A prove can only be obtained in a correctly set up plant-pathogen biotest. My concern here is, that the results cannot be transferred to any agricultural reality. I conclude, that the proposed title of the manuscript is therefore misleading since the article provides no prove that any of the strains may give a benefit in agriculture.

Culture conditions, such as the composition of the growth medium, incubation temperature and time, concentration and method of inoculum preparation, may affect the biosynthesis of bioactive compounds. Therefore, it is important to select an appropriate culture medium when assessing the in vitro antifungal and/or antibacterial activity of the tested strains. For example, the use of a specific carbon source can significantly increase the amount of synthesized secondary metabolites of interest (Bundale et al. 2015; Jakubiec-Krzesniak et al. 2018).

Therefore, the in vitro antibacterial and antifungal activity of Micromonospora isolates is often assessed using different types of culture media and different culture conditions. The confirmation of the antifungal and antibacterial activity of new isolates is a starting point for in vivo analysis and for the search and characterization of new compounds with the potential to be used in human economy.

In vitro testing is also a starting point for further research on the mechanisms of growth promotion by Micromonospora strains and allows selection of strains with positive effects on plant development. As we mentioned in our response to one of the previous comments, the presented results are part of research conducted as part of the doctoral thesis aimed at characterizing plant promotion mechanisms in isolates of the Micromonospora genus and the possibilities of using the most effective strains in agriculture and plant protection. The use of plant endophytes (both bacterial and fungal) that have a beneficial effects on plants is an environmentally friendly approach to promoting plant growth, biological control, increasing stress tolerance, or inhibiting infections caused by phytopathogens (Firdous et al., 2019). Bacteria belonging to the Micromonospora genus are known for their ability to synthesize secondary metabolites, some of which are used in biomedicine. They also play an important role in soil ecology. Recently, due to the protection of the environment and natural resources, the need to limit the use of artificial fertilizers and pesticides is increasingly emphasized. Current practices to support sustainable agriculture are based on the search for new methods to achieve better crop efficiency and environmental protection. One of such strategies is the search and use of microorganisms that positively influence the growth and development of plants, including those related to the protection of plants against phytopathogens. Bacteria of the Micromonospora genus are a good source of biologically active compounds; they also have great potential for use in agriculture as organisms that stimulate the growth and development of plants. Research conducted on new isolates pro-vides information about their potential use in plant protection and biocontrol.

- Bundale, S., Begde, D., Nashikkar, N., Kadam, T., Upadhyay, A. (2015). Optimization of culture conditions for production of bioactive metabolites by Streptomyces spp. isolated from soil. Advances in Microbiology, 5(06), 441. DOI:10.4236/aim.2015.56045.

- Firdous, J., Lathif, N. A., Mona, R., Muhamad, N. (2019). Endophytic bacteria and their potential application in agriculture: a review. Indian Journal of Agricultural Research, 53(1), 1-7. DOI:10.18805/IJARe.A-366.

- Jakubiec-Krzesniak, K., Rajnisz-Mateusiak, A., Guspiel, A., Ziemska, J., Solecka, J. (2018). Secondary metabolites of actinomycetes and their antibacterial, antifungal and antiviral properties. Polish Journal of Microbiology, 67(3), 259-272. DOI:10.21307/pjm-2018-048.

When using mercury for disinfection, how do you handle the waste? Today, we use javel water or simply 70% ethanol.

We agree that there are other sterilizing agents that are less dangerous than mercury chloride. However, this substance is still used to sterilize roots, nodules, and seeds. Mercuric chloride used in a low concentration (0.1%, and this is the concentration we used) has disinfecting properties against soil and epiphytic fungi, which is important when obtaining bacterial isolates (Nurul et al. 2012; Gammoudi et al. 2022).

Diluted mercury chloride solutions are stripped of mercury ions by reducing these ions in a column containing iron dust to metallic mercury. Appropriately marked waste containers are handed over to waste disposal services.

- Nurul, H. D., Shashita, J., Rozi, M. (2012). An improved surface sterilization technique for introducing leaf, nodal and seed explants of Aquilaria malaccensis from field sources into tissue culture. Asia Pac. J. Mol. Biol. Biotechnol, 20, 55-58.

- Gammoudi, N., Nagaz, K., Ferchichi, A. (2022). Establishment of optimized in vitro disinfection protocol of Pistacia vera L. explants mediated a computational approach: multilayer perceptron multiobjective genetic algorithm. BMC Plant Biology, 22(1), 324. DOI:10.1186/s12870-022-03674-x.

The authors have measured the inhibition zones and metallophore-activity zones around the agar plugs. Have the authors compared the zones between strains and between treatments statistically?

The metallophore production activity test and the antifungal properties test were done in triplicates (three biological treatments), whereas the presented values were calculated on the basis of arithmetic mean. Due to the main purpose of the research - the selection of strains with potential growth-promoting properties and evaluation of these potentials in planta and due to the fact that these tests are semi-quantitative, typical statistical analysis like ANOVA test or Kruskal-Wallis test was omitted. Quantitative tests and in-depth analysis will be done for strains whose beneficial effects will be indicated also in vivo or in planta.

Round 2

Reviewer 3 Report

Comments and Suggestions for Authors

Thank you for all these supplementary and important information. I am relieved to lean how you are using mercury and disposing the waste. I understand that mercury is still the "gold-standard" when isolating Actinomycetes.

Your explanations regarding the deployment of your bacteria in an agri- or horticultural setting is not concluding. Your explanations are contradictory.  My rationale is: Micromonospora is a plant endophyte able to produce secondary metabolites with antimicrobial and plant growth promoting activity. This has been shown by other authors. this is not new. Yet, this information is, to my esteem, not sufficient for a publication in the journal Agronomy. Here, it is necessary to go  a step further and study if the isolates are effectively able to suppress disease in a plant-pathogen-Micromonospora setting. 

Another point: the bacterial isolates produce secondary metabolites under certain, well presented, environmental conditions (physical conditions, agar-medium etc.). You argue that you had to find the right medium to make the isolates produce secondary metabolites. I argue, that, in planta, other strains may produce secondary metabolites. It is conceivable that isolates that did not produce any secondary metabolites on agar, do so in planta. From a biocontrol point of view, the ability to colonize plants might be much more important than the production of secondary metabolites. 

Your manuscript describes some strains of a plant endophyte, but it does contibute only very little to biocontrol, as you propose in your introduction. 

Author Response

Dear Reviewer,

According to Your commentaries, we would to explain our assumptions:

'Your explanations regarding the deployment of your bacteria in an agri- or horticultural setting is not concluding. Your explanations are contradictory. My rationale is: Micromonospora is a plant endophyte able to produce secondary metabolites with antimicrobial and plant growth promoting activity. This has been shown by other authors. this is not new. Yet, this information is, to my esteem, not sufficient for a publication in the journal Agronomy. Here, it is necessary to go a step further and study if the isolates are effectively able to suppress disease in a plant-pathogen-Micromonospora setting.'

We agree that the information about the antimicrobial and plant growth promoting potential of Micromonospora members has been evidenced. The data presented in our manuscript confirm previous reports on the beneficial features of these bacteria on the basis of biochemical analysis conducted as part of doctoral research and conclude the first step of this research. According to the research schedule, in planta tests with selected species of plants cultivated in our region and with plants known as hyperaccumulators of heavy metals and metalloids will be carried out in the next step, where the most promising strains of bacteria will be used for inoculation of the  tested plants growing in various conditions. Additionally, the in planta confirmed ability to promote plant growth or to inhibit phytopathogenic fungi will be analyzed to show the mechanisms of action, especially because various Micromonospora strains can produce different antimicrobial substances including antibiotics, even not yet discovered, but whose catalog is constantly expanding and offers new possibilities for their applications (Zhao et al., 2017; Kaari 2023). Due to the restrictions related to the cultivation of plants infected with phytopathogens, we have contacted and plan to conduct such research in a center with appropriate permits and a place intended for such cultivation (Institute of Horticulture - National Research Institute Skierniewice). It should be emphasized that the reports about the new strains with potential to be used in plant protection or plant nutrition are in our opinion interesting information due to the necessity of finding a non-chemical alternative for commonly used chemical pesticides (Maitra et al., 2021).

Kaari, M., Manikkam, R., Annamalai, K. K., Joseph, J. (2023). Actinobacteria as a source of biofertilizer/biocontrol agents for bio-organic agriculture. Journal of Applied Microbiology, 134(2), lxac047.

Maitra, S., Brestic, M., Bhadra, P., Shankar, T., Praharaj, S., Palai, J. B., (...), Hossain, A. (2021). Bioinoculants-natural biological resources for sustainable plant production. Microorgan-isms, 10(1), 51.

Zhao, S., Liu, C., Zheng, W., Ma, Z., Cao, T., Zhao, J., (…), Wang, X. (2017). Micromono-spora parathelypteridis sp. nov., an endophytic actinomycete with antifungal activity isolated from the root of Parathelypteris beddomei (Bak.) Ching. International Journal of Systematic and Evolutionary Microbiology, 67(2), 268-274.

'Another point: the bacterial isolates produce secondary metabolites under certain, well presented, environmental conditions (physical conditions, agar-medium etc.). You argue that you had to find the right medium to make the isolates produce secondary metabolites. I argue, that, in planta, other strains may produce secondary metabolites. It is conceivable that isolates that did not produce any secondary metabolites on agar, do so in planta. From a biocontrol point of view, the ability to colonize plants might be much more important than the production of secondary metabolites.'

We agree that the antifungal activity of bacteria is dependent on many biotic and abiotic factors, i.e. environmental conditions (in vitro or in planta), physical conditions, or agar-medium composition. In our manuscript, we report on the antifungal ability of bacteria against common phytopathogenic fungi, which was shown on the basis of the in vitro evaluation. This antifungal ability is strongly dependent on the carbon and nitrogen sources. In future, we will try to isolate and indicate the substance/substances responsible for antifungal activity. Moreover, the appropriate methods to produce antifungal substances may be used in commercial production of bioinoculants (Vassileva et al., 2021; Silva et al., 2022).

Silva, G. D. C., Kitano, I. T., Ribeiro, I. A. D. F., Lacava, P. T. (2022). The potential use of actinomycetes as microbial inoculants and biopesticides in agriculture. Frontiers in Soil Science, 2.

Vassileva, M., Malusà, E., Sas-Paszt, L., Trzcinski, P., Galvez, A., Flor-Peregrin, E., (...), Vassilev, N. (2021). Fermentation strategies to improve soil bio-inoculant production and quali-ty. Microorganisms, 9(6), 1254.

'Your manuscript describes some strains of a plant endophyte, but it does contibute only very little to biocontrol, as you propose in your introduction.'

We agree that some statements included in our manuscript suggest that our strains have potential as biocontrol factors. These sentences have been removed or changed as follows:

The title of the manuscript: ‘In vitro screening of endophytic Micromonospora strains associated with white clover for biological control of phytopathogenic fungi and promotion of plant growth’ was corrected. Phrase ‘biological control’ has been deleted and now the title reads: ‘In vitro screening of endophytic Micromonospora strains associated with white clover for antimicrobial activity against phytopathogenic fungi and promotion of plant growth’.

Phrase: ‘To evaluate their potential as biocontrol agents’ (lines 48-49) has been removed.

Sentence: ‘The results indicate the in vitro potential of selected strains as biocontrol factors and plant growth promotion.’ (line 295-298) has been changed. The information about the necessity of deeper analysis of bacterial biocontrol activity has been added: ‘The results indicate the in vitro potential of selected strains as plant growth promotion agents. Further studies are necessary to evaluate the use of the studied strains in biocontrol of phytopathogenic fungi’.